# CNBP Binds and Unfolds In Vitro G-Quadruplexes Formed in the SARS-CoV-2 Positive and Negative Genome Strands

**DOI:** 10.3390/ijms22052614

**Published:** 2021-03-05

**Authors:** Georgina Bezzi, Ernesto J. Piga, Andrés Binolfi, Pablo Armas

**Affiliations:** 1Instituto de Biología Molecular y Celular de Rosario (IBR), Consejo Nacional de Investigaciones Científicas y Técnicas (CONICET), Universidad Nacional de Rosario (UNR), Ocampo y Esmeralda, Rosario S200EZP, Santa Fe, Argentina; bezzi@ibr-conicet.gov.ar (G.B.); piga@ibr-conicet.gov.ar (E.J.P.); binolfi@ibr-conicet.gov.ar (A.B.); 2Plataforma Argentina de Biología Estructural y Metabolómica (PLABEM), Ocampo y Esmeralda, Rosario S200EZP, Santa Fe, Argentina

**Keywords:** COVID-19, SARS-CoV-2, coronavirus, G-quadruplex, CNBP

## Abstract

The Coronavirus Disease 2019 (COVID-19) pandemic has become a global health emergency with no effective medical treatment and with incipient vaccines. It is caused by a new positive-sense RNA virus called severe acute respiratory syndrome-related coronavirus 2 (SARS-CoV-2). G-quadruplexes (G4s) are nucleic acid secondary structures involved in the control of a variety of biological processes including viral replication. Using several G4 prediction tools, we identified highly putative G4 sequences (PQSs) within the positive-sense (+gRNA) and negative-sense (−gRNA) RNA strands of SARS-CoV-2 conserved in related betacoronaviruses. By using multiple biophysical techniques, we confirmed the formation of two G4s in the +gRNA and provide the first evidence of G4 formation by two PQSs in the −gRNA of SARS-CoV-2. Finally, biophysical and molecular approaches were used to demonstrate for the first time that CNBP, the main human cellular protein bound to SARS-CoV-2 RNA genome, binds and promotes the unfolding of G4s formed by both strands of SARS-CoV-2 RNA genome. Our results suggest that G4s found in SARS-CoV-2 RNA genome and its negative-sense replicative intermediates, as well as the cellular proteins that interact with them, are relevant factors for viral genes expression and replication cycle, and may constitute interesting targets for antiviral drugs development.

## 1. Introduction

By the end of 2019, an unexpected outbreak of a new severe acute respiratory syndrome (SARS) termed by the World Health Organization (WHO) as Coronavirus Disease 2019 (COVID-19) emerged in Wuhan (China) [1]. The cause was infection by a highly contagious new SARS-related coronavirus named SARS-CoV-2 [2], which rapidly spread around the world. On 11 March 2020, WHO declared COVID-19 a “pandemic” condition. In one year, the global number of confirmed infections was nearly 90 million and the death toll over 1.9 millon, which is spread over more than 200 countries and with repetitive and constantly increasing waves of contagion (https://covid19.who.int/, accessed on 14 January 2021). Despite the enormous efforts of the medical and scientific community and the pharmaceutical industry, so far no clearly effective pharmacological treatments have been found for treating SARS-CoV-2 infection and disease, and the first immunizations with vaccines developed in record times are beginning. Therefore, there is urgency for the development of specific drugs and novel treatments.

SARS-CoV-2 is a betacoronavirus genus of the *Coronaviridae* family, and is one of the seven types of the *Coronaviridae* family of viruses which could infect humans. Four of them, two alphacoronaviruses (HCoV-229E and HCoV-NL63) and two betacoronaviruses (HCoV-HKU1 and HCoV-OC43), are common around the world and cause mild diseases [3]. The other three human betacoronaviruses are more recent, causing severe acute respiratory outbreaks. SARS-CoV emerged in 2002 and 2003 in Guangdong province (China) [4]. The Middle East Respiratory Syndrome Coronavirus (MERS-CoV) was identified in Saudi Arabia in 2012 [5], and in 2019 SARS-CoV-2 caused the COVID-19 pandemic. SARS-CoV and MERS-CoV originated from bats, and it appears to be so for SARS-CoV-2 as well, which shows fairly close relatedness with three bat-derived coronaviruses, bat-SL-CoVZC45 and bat-SL-CoVZXC21 [6], as well as RaTG13 [1]. SARS-CoV is the most genetically related to SARS-CoV-2 among human coronaviruses [6], sharing a high nucleotide sequence identity (79.7%). However, the replication rate of SARS-CoV-2 is higher than that of SARS-CoV [7]. All these viruses are enveloped viruses with positive-sense single-stranded RNA genomes of about 30 kb in length and have similar structures, genomic organizations, and replicative cycles [3,8]. Upon infection of a host cell, the positive-sense RNA genome (+gRNA) is released and ready to be translated by the protein synthesis machinery of the infected host cell to express a set of viral proteins crucial for viral replication [9]. Both, replication of the viral genome and transcription of positive-sense subgenomic RNAs (+sgRNAs) involve the synthesis of negative-strand RNA genome (−gRNA) and negative-strand subgenomic RNAs (−sgRNAs) intermediates [10]. As other RNA viruses, SARS-CoV-2 is dependent on effectively engaging host cell factors such as regulators of RNA stability, processing, localization, and translation to facilitate replication and production of new viral particles [11]. On the other hand, the infected host cell must detect the pathogen and activate appropriate innate immune response pathways to restrict virus infection [11]. Recent comprehensive studies have begun to identify expression changes or modifications in the host cell transcriptome [12,13] and proteome [14,15,16] as well as cellular proteins interacting with viral proteins [17,18] and with viral genome [19], as approaches to identify cellular pathways relevant for viral infection and replication. However, a more detailed understanding of the molecular mechanisms and interactions occurring during SARS-CoV-2 infection is required to design efficient therapeutic strategies.

Viral RNA genomes have some intrinsic characteristics that favor or obstruct the viral genome expression and replication. For instance, the folding of specific regions of the genomic RNA molecule into stable secondary structures may act as specific hallmarks for the attachment of cellular or viral RNA processing machinery, but may also be roadblocks for viral RNA metabolism [20]. Among these structures, G-quadruplexes (G4s) are stable four-stranded structures formed in G-rich DNA or RNA sequences that can be formed by the folding on itself of a single-stranded molecule [21]. The structure is characterized by the stacking of two or more planar arranges of four G nucleobases (called G-tetrads) stabilized by lateral Hoogsteen-type hydrogen bonds and by the coordination of monovalent cations, mainly K^+^ (Figure 1a). These structures may occur in putative G-quadruplex sequences (PQSs) presenting at least four contiguous tracts of two or more guanine nucleotides interspersed with short nucleotide sequences forming the G4 loops. Depending on the relative orientation of the G-tracts, G4s may be parallel (with four G-tracts in the same relative orientation), antiparallel (with two G-tracts in opposite orientation in respect to the other two), or hybrid (with one G-tract in opposite orientation in respect to the other three). G4s have received extensive attention during the last two decades due to their involvement in the regulation of cellular processes such as transcription, replication, translation, and telomere maintenance and the development of specific G4 ligands with promising anticancer effects [21].

Besides in human beings, G4s are widespread across nucleic acids of all the taxonomic phyla [22] including *Bacteria* [23], *Archaea* [24], and viruses [25,26]. Critical roles for viral G4s have been described in human viruses, including immunodeficiency virus (HIV), herpes simplex virus (HSV), Epstein-Barr virus (EBV), human papillomavirus (HPV), hepatitis B virus (HBV), Nipah virus, hepatitis C virus (HCV), Zika virus, and Ebola virus [27,28,29], and some G4-specific compounds have shown powerful antiviral activity by targeting G4 structures [28,29]. Very recent reports have initiated the search for G4s in the genomes of human coronaviruses, including SARS-CoV-2 [30,31,32,33,34,35,36]. Beyond the prediction of PQSs, some works have demonstrated the formation of a few G4s in vitro by PQSs found in the +gRNA [30,34,35,36] and one of them was demonstrated to be formed within cultured human cells and controls the translation efficiency of the nucleocapsid N protein [35,36].

Here, we have used a novel G4 prediction pipeline for the identification of the PQSs with high probability of G4 formation within the +gRNA and −gRNA of SARS-CoV-2 and related betacoronaviruses. We also performed a predictive analysis of the putative consequence of natural nucleotide variations reported for SARS-CoV-2 on the probability of G4 formation in the selected PQSs, showing that some synonymous mutations may alter G4 formation with putative consequences in their regulatory functions. Then, using multiple biophysical techniques, we confirmed the formation of two G4s in the +gRNA and provide the first evidence of G4 formation by two PQSs in the −gRNA of SARS-CoV-2. Finally, we performed biophysical and molecular approaches to demonstrate for the first time that cellular nucleic acid binding protein (CNBP), the main cellular protein bound to SARS-CoV-2 RNA genome in infected human cells [19], binds and promotes the unfolding of G4s formed by both +gRNA and −gRNA of SARS-CoV-2. Our results suggest that G4s found in SARS-CoV-2 +gRNA and −gRNA, and the cellular proteins that interact with them, are important elements for viral replication cycle and may be novel targets for developing antiviral drugs against COVID-19.

## 2. Results and Discussion

### 2.1. G4 Prediction and Selection in the SARS-CoV-2 Positive and Negative Genome

During the COVID-19 pandemic, mainly along the last half of 2020, several works analyzed the SARS-CoV-2 RNA genome to seek PQSs. All of them analyzed the +gRNA [30,31,32,33,34,35,36], while a few made a superficial overview of PQSs on the −gRNA [32,33,36]. Although −gRNA and −sgRNAs are minority in respect of their positive-sense RNA counterparts and represent only about 1% of viral RNA [9,10], negative-sense RNAs are key intermediates functioning as templates for +gRNA replication and +sgRNAs transcription. These processes are mediated by the replicase-transcriptase complex (RTC) formed by several non-structural proteins (nsps) [9,10]. Consequently, −gRNA and −sgRNAs may contain G4s with putative regulative functions on these processes. In addition, most of the previous PQSs analyses on SARS-CoV-2 genome have used different bioinformatics prediction tools, some of them designed for DNA-G4, and a variety of criteria for selecting the best PQS candidates, mainly including higher prediction scores [30,31,32,33,34,35,36] combined with lower potential thermodynamic stability of secondary structures competitive with G4s [32], uniqueness in SARS-CoV-2 and conservation among variants of SARS-CoV-2 [34], and conservation among human coronaviruses [36]. Although there is divergence in G4 prediction tools and selection criteria, none of the predictions have found PQSs with four tracts of three consecutive guanines (with the potential of forming three-tetrads G4s), and have only found PQSs with four tracts of two consecutive guanines (with the potential of forming two-tetrads G4s). Although two-tetrads G4s are less stable than the three-tetrads G4s, especially in vivo, it is known that the RNA G4s are more stable than their DNA counterparts [37] and several emerging studies have demonstrated the formation of two-tetrads G4s in viral sequences [38,39,40,41,42,43,44].

Here, we have performed a predictive analysis of PQSs on the +gRNA and −gRNA sequences from three coronaviruses that infect humans and cause the most severe health consequences: the SARS-CoV-2, SARS-CoV (or SARS-CoV-1), and MERS-CoV. We also included genomes from three bat coronaviruses probably related to SARS-CoV-2 origin: bat-SL-CoVZC45 and bat-SL-CoVZXC21 [6], as well as RaTG13, which presents a closer phylogenetic relationship with SARS-CoV-2 [1]. First, we downloaded the +gRNA sequences of the analyzed viruses and obtained for each one the respective reverse complement sequence for −gRNA analysis. Then, we performed an initial analysis of the PQSs found in the +gRNA and −gRNA obtained by using Quadruplex forming G-Rich Sequences (QGRS) Mapper online prediction software [45] for the identification of canonical two-tetrads PQSs (with four two-guanine tracts) and loops of extended lengths (from 1 to 15 nucleotides), i.e., G_2+_N_1–15_G_2+_N_1–15_G_2+_N_1–15_G_2+_. The retrieved PQSs were then analyzed using two additional predictors: PQSfinder Web and G4RNA screener. PQSfinder Web [46] is an algorithm that supports DNA and RNA sequences but was validated primarily on DNA sequences and has been trained with G4-seq data. G4RNA screener [47] is a web algorithm that identifies regions in RNA sequences prone to fold into G4 based on three scoring systems: cGcC (Consecutive G over consecutive C ratio) [48], G4H (G4Hunter) [49], and G4NN (G4 Neural Network) [50]. Results from this analysis are detailed in Appendix A (for +gRNA, each virus in a different tab) and Appendix A (for −gRNA, each virus in a different tab). Based on the scores obtained using the five G4 predictors, we defined the following selection criterion: PQSs that were found with QGRS Mapper and which scores for the other four predictors were over the defined threshold for each predictor were classified with high probability to form G4 (highlighted in red), those PQSs that were found with QGRS Mapper and which scores for at least three of other four predictors were over the defined threshold for each predictor with at least one of those scores significantly high were classified with medium probability to form G4 (highlighted in orange), and those PQSs that were found with QGRS Mapper and which scores for at least three of other four predictors were over the defined threshold for each predictor with none of those scores significantly high were classified with low probability to form G4 (highlighted in yellow). In addition, we highlighted those PQSs that present significantly high cGcC scores (marked with thick edges), although some of them did not fulfill the selection criterion. A summary of the bioinformatic workflow is presented in Figure 1b, while a summary of the numbers of this analysis is represented in Table 1 and a schematic location of the selected PQSs on the explored viral genomes is shown in Figure 1c.

Our results show that PQSs predicted by QGRS Mapper are scattered along the genomes of the five analyzed viruses showing 29 to 49 PQSs in the +gRNA and 16 to 38 PQSs in the −gRNA (Table 1). SARS-CoV-2 and RaTG13 present an intermediate number of PQSs in both strands, with 37 PQSs in the +gRNA and 19 PQSs in the −gRNA of SARS-CoV-2 while 20 PQSs in the −gRNA of RaTG13. SARS-CoV is the one that displays the highest amount of initial PQSs predicted for the +gRNA followed by MERS-CoV and SARS-CoV-2, while bat-SL-CoVZC45 and bat-SL-CoVZXC21 show the lowest amount of initial PQSs predicted in the +gRNA. A similar order is observed for the number of initial PQSs predicted for the −gRNA, except for the fact that MERS-CoV presents the higher number of initial PQSs followed by SARS-CoV, SARS-CoV-2 and RaTG13, while bat-SL-CoVZC45 and bat-SL-CoVZXC21 again show the lowest amount of initial PQSs predicted in the −gRNA. The numbers of PQSs predicted by QGRS Mapper partially correlate with genomes G content, being the genomes of SARS-CoV and MERS-CoV which show the highest G% in both + and −gRNA and are the ones that present the highest numbers of PQSs, while the genomes of bat-SL-CoVZC45 and bat-SL-CoVZXC21 show the lowest G% in both + and −gRNA and are the ones that present the lowest numbers of PQSs. Curiously, SARS-CoV-2 and RaTG13 show the lowest G% in both + and −gRNA, but even so shows higher numbers of PQSs than the genomes of bat-SL-CoVZC45 and bat-SL-CoVZXC21, probably indicating that SARS-CoV-2 and RaTG13 may have gained PQSs during their evolution from the putative common ancestor shared with bat-SL-CoVZC45 and bat-SL-CoVZXC21. In agreement with this, the lower number of PQSs and lower G% found in SARS-CoV-2 compared to SARS-CoV may be related to the fact that SARS-CoV-2 replicates faster than SARS-CoV because G4 structures may represent an obstacle for viral proteins translation and RNA dependent RNA synthesis [36]. Noteworthy, the numbers of selected PQSs by our criterion using several predictors show a clear difference from the numbers of initial PQSs predicted by QGRS Mapper. SARS-CoV-2 presents the lowest number of selected PQSs in the +gRNA (only ≈8% of the PQSs originally predicted), probably indicating a negative selection of PQSs capable of forming stable G4s in this virus. This is in agreement with the previously reported data indicating that SARS-CoV-2 displays general PQSs poverty when compared to the virus realm, its PQS density being in the lower end of results from the *Coronaviridae* family, which itself is in the lower end of the (+) ssRNA Group IV [34] and the PQS frequency in SARS-CoV-2 is significantly lower than expected from its base composition [33]. On the contrary, the SARS-CoV-2 −gRNA presents the highest percentage of selected PQSs from the initially predicted PQSs by QGRS Mapper (≈26%), and a similar tendency is observed for bat-SL-CoVZC45, bat-SL-CoVZXC21, and MERS-CoV, while a lower percentage is observed for RaTG13 and SARS-CoV. This may indicate a positive selection of PQSs capable of forming stable G4s in the −gRNA of SARS-CoV-2 with potential regulatory functions in replication/transcription. This fact may be the consequence that −gRNA and −sgRNAs are not templates for translation and their evolution may not be constrained by the negative effect of the G4s on viral proteins translation.

Visual analysis of the location of the selected PQSs on the explored viral genomes (Figure 1c) shows that SARS-CoV-2 and bat coronaviruses display a higher PQS density in the genome region coding for the nsps (encoded by ORF1ab) and lower PQS density in the genome region coding for the structural proteins. Based on the location of the selected PQSs, we identified six of them (three from the +gRNA and three from the −gRNA) that are conserved in position in at least four viral genomes and analyzed their sequence conservation (Appendix A). From the six selected PQSs conserved in position, five of them (three from the +gRNA and two from the −gRNA) show very high sequence conservation among SARS-CoV-2 and bat coronaviruses (>88% except for +28,880 PQS of RaTG13) and lower identity % with SARS-CoV and MERS-CoV (when they had PQSs conserved in position), while one selected PQS conserved in position from the −gRNA is neither conserved among SARS-CoV-2 and bat coronaviruses nor with SARS-CoV and MERS-CoV. It is noticeable that one of the PQSs that presented significantly high cGcC scores but did not fulfill our selection criterion (position +13,385 in SARS-CoV-2 +gRNA, see Appendix A) is conserved in position and sequence among the six viral genomes studied (Figure 1c and Appendix A) and has been previously shown to fold in vitro as G4 [30,35]. Interestingly, this PQS is located very near (≈80 nucleotides upstream) the slippery sequence that causes the ribosomal frameshift that controls the transition from the translation of the ORF1a to the translation of the ORF1ab, making it an attractive PQS forming a G4 with putative function in the regulation of this process together with the pseudoknot structure already described [51].

In this work, we focused the following experimental studies on the SARS-CoV-2 PQSs highly conserved among the explored virus genomes. Conservation in PQSs candidates is a trace of maintenance through natural selection and indicates that selected PQSs may be relevant elements for the biological fitness of these viruses, beyond SARS-CoV-2 and extended to those viruses that share the conserved PQSs. Table 2 shows the main characteristics of the five selected PQSs with conserved positions and sequences among at least four of the explored coronaviruses. All of the selected PQSs had been previously predicted using different strategies and predictors, and for those in the +gRNA (+644, +3467 and +28,903) there are experimental evidences of G4 formation. However, until now, no experimental analysis of G4 formation has been performed for PQSs predicted in the −gRNA. 

Finally, we performed an analysis of variations within these PQSs using GISAID database (https://www.gisaid.org/, accessed on 2 January 2021) [52]. Table 2 shows the reported variations and highlights those that may lead to impede PQSs and those that produce G tracts extension (and probably higher propensity to form stable G4s). Appendix A contains further information about the GISAID mutations found in the selected PQSs, including frequency, codon, protein, amino acid change, and scores for the PQSs predictors used in this work. Of note, all the analyzed nucleotidic changes show very low frequencies (<1%) and most of them (28/48) disrupt G-tracts and may lead to impede PQSs or reduce PQSs scores, while only 8/46 produce G-tracts extension and may increase PQSs scores, the other 10/46 changes being those that do not disturb G-tracts and may be neutral for PQSs (although some of them present variations in scores which may lead to G4 stabilization or destabilizations). The PQS that showed the higher number of variations is +28,903, mainly of the G4-disruptive type (16/20) and none of the G4-stabilizing type. Interestingly, all the G4-stabilizing variations are synonymous or silent mutations with no consequence in the encoded amino acids, while most of the G4-disruptive variations are not synonymous (missense or frameshift) mutations producing changes in the encoded amino acids (27/28). Considering that single-nucleotide and short variations in PQSs may affect G4s formation or stability with consequences in transcriptional [53,54,55] and translational [56,57,58] control, it would be important to analyze the mutations occurring within PQSs not only for their effects on encoded proteins, but also for the putative effects in G4-regulated processes.

On the other hand, non-conserved PQSs that are unique for a particular virus may also play a central role in the ability of the virus to adapt to new environmental challenges and infect and replicate in novel hosts. This could be the case of SARS-CoV-2 PQSs of the −gRNA in positions −25,003 (which is unique for this virus) and −13,134 (which is only conserved in RaTG13 genome, Figure 1c and Appendix A) or the one in position −19,865, whose sequence is not fully conserved (except in RaTG13 genome, which shows a higher conservation but the PQS in this position did not fulfill our selection criterion, Figure 1c and Appendix A). None of these PQSs were selected for further study in this work, but they remain as interesting candidates to study SARS-CoV-2 specific G4s.

Although many of the identified PQSs were previously described by other approaches, our selection criterion has highlighted some new PQSs, mainly those in the −gRNA, as interesting candidates to perform experimental studies.

### 2.2. Confirmation That the Selected PQSs Fold In Vitro as G4

The five selected PQSs were further studied in their capability to form G4 structures in vitro using synthetic RNA oligoribonucleotides for four different spectroscopic approaches: Circular Dichroism (CD) Spectroscopy (Figure 2a and Appendix A), 1D ^1^H Nuclear Magnetic Resonance (NMR) (Figure 2b), Thermal Difference Spectroscopy (TDS) (Appendix A), and Thioflavin T (ThT) fluorescence (Appendix A).

For PQSs +3467 and −23,877, CD spectra have the typical pattern of peaks associated with parallel G4 structure, showing an increase of a positive peak around 263 nm and a negative peak around 240 nm in response to the presence of increasing K^+^ concentrations (Figure 2a). K^+^ is considered the main intracellular G4-stabilizing cation [59] and the CD positive peaks of these two PQSs easily reached the maximum intensity with K^+^ concentrations above 10 mM. The characteristic G4 spectra were not observed in the presence of Li^+^, which plays a neutral role in G4 folding and stability [59]. CD melting curves (Appendix A) showed that these G4s are stable structures with estimated Tm of 58.5 °C (for PQSs +3467) and 51.6 °C (for PQS −23,877) and high values of ΔG, indicative of high stabilities for two-tetrads G4s. In addition, 1D ^1^H NMR showed defined signals around 11–12 ppm (Figure 2b), confirming the presence of Hoogsteen bonds and G4 structures. In agreement with the former results, TDS spectra showed the typical G4 signature with two positive peaks around 243 and 273 nm and a negative peak at 295 nm (Appendix A), and ThT fluorescence assays showed that these folded PQSs notably enhance ThT fluorescence above 30-fold for +3467 and above 50-fold for −23,877 (Appendix A). In coincidence, the prediction of the secondary structures of these PQSs by RNAfold predicts the G4 structures (at 20 °C) and NUPACK and RNAfold software do not predict stable secondary structures that may compete with G4 formation (Appendix A).

In the case of PQSs +644 and −13,963, CD spectra also showed the peaks associated with parallel G4s, but with a milder increase in the positive peak with the increase of K^+^, which was not observed with Li^+^ (Figure 2a). For these two PQSs, high K^+^ concentrations (up to 200 mM) were needed so as to observe a clear G4 spectra, probably indicating that these G4s are less stable or less prone to fold. CD melting (Appendix A) showed that these G4s are less stable than PQSs +3467 and −23,877, showing estimated Tm of 52 °C (for PQS +644) and 48.5 °C (for PQS −13,963) and ΔG values slightly lower than those for PQSs +3467 and −23,877. 1D ^1^H NMR also showed G4 signatures but signals were less intense than those for +3467 and −23,877 (Figure 2b), suggesting that there is a less amount of G4 probably due to lower stabilities and loose global conformation. TDS spectra showed the typical G4 signatures for the PQS +644 and a less defined spectrum for the PQS −13,963 (Appendix A), while ThT fluorescence assay showed that both PQSs increase ThT fluorescence barely above 10-fold (Appendix A). For these PQSs, RNAfold does not predict G4 structures and both, RNAfold and NUPACK, do not predict stable secondary structures for +644 and predict a weak stem-loop structure with three base pairs for −13,963 (Appendix A), indicating that the G4s might not compete with stable alternative structures.

PQS +28,903 showed CD spectra with a positive peak centered at 270 nm that remain unperturbed upon K^+^ additions, even reaching K^+^ concentration of 200 mM (Figure 2a). In addition, similar spectra were observed in the absence of added monovalent cation or in the presence of Li^+^, suggesting that this sequence do not adopt a G4 structure. CD melting (Appendix A) showed that the observed structure is one of the less stable ones, with an estimated Tm = 49.3 °C and the lowest calculated ΔG value. 1D ^1^H NMR did not show G4 signals in the 11–12 ppm region but showed a clear signal around 13.7 ppm (Figure 2b), indicating that this sequence does not form Hoogsteen bonds (i.e., G4 structures) but instead contains Watson-Crick base pairs at some extent [60]. TDS spectrum displayed a very low signal and poor defined spectrum (Appendix A), which further supports the absence of G4 and is probably compatible with self-complementary duplex structure [61] and ThT fluorescence assay showed that this PQS does not significantly increase ThT fluorescence (only 3-fold) (Appendix A). In coincidence, although RNAfold and NUPACK do not predict intermolecular duplex (self-dimers), they predict a relatively stable stem-loop structure with four base pairs for this PQS (Appendix A), which may compete with G4 formation and may account for the signatures observed in NMR and TDS spectra.

Overall, our data showed that, except for the PQS +28,903, the other selected PQSs form G4 structures, +3467 and −23,877 being the ones with higher stability and/or propensity to form, followed by +644 and −13,963. In agreement with our results, the PQS +644 has been reported to fold as G4 in vitro by using Thioflavin T (ThT) fluorescence assay and CD [36]. Similarly, the PQS +3467 has been reported to fold as G4 in vitro by using CD and 1D ^1^H NMR [34], showing very similar spectra for both methods. Surprisingly, the PQS +28,903 was also reported to fold as G4 in vitro by several works [34,35,36], not only by N-methyl mesoporphyrin IX (NMM) and ThT fluorescence assays, CD and 1D ^1^H NMR, but also by native PAGE mobility assays, fluorescence resonance energy transfer (FRET) combined with stopped flow, and PCR-stop assays in combination with PQS mutations and G4 stabilizing ligands. In addition, the PQS +28,903 was also informed to be formed within living cells, where it is capable of inhibiting the translation of a reporter gene (GFP) [36] and of the SARS-CoV-2 N protein [35] upon incubation with G4 stabilizing ligands, positioning this PQS as an interesting target for the design of SARS-CoV-2 antiviral strategies. In our experimental conditions, this PQS was not able to form a defined and stable G4, although it formed a stable secondary structure containing Watson-Crick bonds, as was evident in the ^1^H NMR spectrum. Interestingly, in previously reported NMR spectra, similar Watson-Crick bonds peaks were observed around 13 ppm [34]. In summary, our results confirm the formation of G4 by two PQS found in SARS-CoV-2 +gRNA (i.e., +644 and +3467), and inform for the first time the formation of G4 by two PQSs found in SARS-CoV-2 −gRNA (i.e., −13,963 and −23,877), while they could not confirm the folding as G4 structure of the PQS +28,903 found in SARS-CoV-2 +gRNA. This suggests that other PQSs than the +28,903 may also be interesting targets for testing their biological role and antiviral strategies specific for G4.

### 2.3. Cellular Nucleic Acids Binding Protein (CNBP) Interacts with Some SARS-CoV-2 G4s and Promotes Their Unfolding

Viral reproduction depends at some points on host cellular machinery. The antiviral strategies that target viral proteins are usually effective only against specific viral strains and fails even for closely related viral species or mutant virus from the same species. However, targeting host proteins needed for viral replication cycle is a better strategy to achieve a wide range response toward viruses that make use of common cellular pathways. This is why, since the COVID-19 pandemic outbreak, several scientific groups around the world have made efforts to describe human cellular proteins interacting with SARS-CoV-2 viral components, not only to better understand the mechanism of viral infection and the host innate immune response, but also to discover new targets for antiviral therapy. The first studies on SARS-CoV-2-infected human cells have focused on characterizing changes in the host cell transcriptome [12,13] or proteome [14,15,16] and interactions between viral proteins and host proteins [17,18], revealing cellular pathways relevant to productive infection. However, these studies could not reveal how viral RNA is regulated during infection or how viral infection remodels host cell RNA metabolism to enable its replication. A bioinformatic approach has recently predicted human RNA-binding proteins sites in SARS-CoV-2 RNA proposing three highly promising candidates (SRSF7, HNRNPA1, and TRA2A) that are involved in cellular RNA metabolism and share multiple RGG-rich novel interesting quadruplex interaction (NIQI) motifs common to most G4 binding proteins [33]. A more recent work has identified 104 human proteins that directly and specifically bind to SARS-CoV-2 RNAs in infected human cells by using RNA antisense purification and quantitative mass spectrometry (RAP–MS) [19]. Among the identified cellular proteins, CNBP, also referred to as zinc finger protein 9 (ZNF9), was the human protein most significantly enriched in RAP–MS. CNBP was even more enriched than the 15 viral proteins found in the same study, which comprised of 5 structural proteins (included the N, S and M proteins) and 10 nsps known to bind viral RNA. Moreover, CNBP was strongly upregulated in a proteome analysis of human cells after viral infection [19], and, among all SARS-CoV-2 RNA interactome proteins, CNBP had the most significant effect on virus-induced cell death in a genome-wide CRISPR perturbation screen designed to identify host factors that affect cell survival after SARS-CoV-2 infection [62]. CNBP is a highly conserved nuclear-cytoplasmic protein with nucleic acid chaperone activity [63] that preferentially binds to G-enriched RNA or DNA single-stranded sequences [64,65]. CNBP contains an RGG-rich NIQI motif and has been recently described as a transcriptional regulator that unfolds G4 in the promoters of *c-MYC* and *KRAS* oncogenes and in the *NOG* developmental gene [66]. On the other hand, CNBP has been also reported to boost global translation by resolving G4 structures in the 5′ UTRs of mRNAs [65]. Considering this scenario, we decided to assay the binding and function of CNBP on the G4s identified in this work.

Electrophoretic mobility shift assays (EMSAs) were performed to evaluate CNBP capability to bind to the PQSs folded in the presence of K^+^ or Li^+^ (Figure 3a). PQSs +644, +3467 and −23,877 clearly interacted with CNBP, as evidenced by a shift of the PQSs mobilities. Binding of CNBP to PQS −13,963 was less evident, as only mild shifts were observed at high CNBP concentrations. Instead, the PQS +28,903 did not interact with CNBP in our experimental conditions. For the PQSs that showed the best interaction (i.e., +644, +3467 and −23,877), a slightly higher affinity was observed in the condition folded in presence of Li^+^ than in the condition folded in presence of K^+^, since in the Li^+^ condition shift is observed (and/or free probes are consumed) at lower CNBP concentrations. This is in agreement with a previous report proposing that CNBP promotes G4s unfolding by shifting the equilibrium between G4 and unfolded (single-stranded) states towards the unfolded state through preferentially binding to the unfolded sequence, thus avoiding G4 re-folding [66]. To evaluate CNBP ability to unfold SARS-CoV-2 G4s studied in this work, we performed CD spectra of previously folded G4 incubated with CNBP or with BSA as an unspecific protein with no G4 unfolding activity (Figure 3b). CNBP caused a distortion and reduction of the characteristic G4 peaks of the spectra of the PQSs +644, +3467, −13,963 and −23,877, but it did not significantly affect the spectrum of the PQS +28,903. BSA did not affect any of the CD spectra. These results indicate that CNBP is capable of binding in vitro to the PQSs +644, +3467, −13,963 and −23,877 and unfolding the preexistent G4s that they form. Instead, the PQS +28,903 did not interact with CNBP and the structure detected by CD was not significantly affected by the protein.

Among several diverse functions assigned to CNBP, it was reported to induce the transcription of sustained pro-inflammatory cytokines by binding to specific short sequences in their promoters and activate its own transcription in a positive feedback mechanism of autoregulation in response to stimulation with lipopolysaccharide [67]. CNBP also induces IL-12β (Il12b) mRNA synthesis in response to diverse microbial pathogens that engage multiple pattern recognition receptors [68]. *Cnbp*-deficient mice fail to mount protective IL-12 and IFN-γ responses in vivo, resulting in a reduced Th1 cell immune response and an inability to control parasite replication [68]. Based on these data, CNBP has been suggested as a key transcriptional regulator required for activating and maintaining the immune response. These findings are consistent with CNBP-depleted cells being sensitized to virus-induced cell death [19], which suggests that CNBP may act as an antiviral regulator. Enhanced crosslinking and immunoprecipitation (eCLIP) in SARS-CoV-2-infected cells has shown CNBP binding along SARS-CoV-2 +gRNA with several strongly enriched binding sites [19]. However, it remains to be determined if CNBP role on SARS-CoV-2 replication is due to its action on viral RNA or to its action on cytokine and other cellular genes expression regulation. Here, we provide evidence that CNBP is capable of interacting and unfolding SARS-CoV-2 G4s in both the +gRNA and the −gRNA with putative regulative functions in SARS-CoV-2 gene expression and replication.

### 2.4. Integration of Results with Putative Functions of G4 and CNBP in SARS-CoV-2 Replicative Cycle

Replication cycle of the SARS-CoV-2 (Figure 4) initiates with the binding of the virus to the host cell by interaction of the S protein with its receptor, the angiotensin-converting enzyme 2 (ACE2) (Figure 4, step 1). Following receptor binding, the virus enters host cell by endocytosis and then to the cytosol via proteolytic cleavage of S protein, followed by fusion of the viral and cellular membranes (Figure 4, step 2). After the +gRNA enters the host cell, it is translated by cytosolic cellular ribosomes to synthesize the viral components of RTC. RTC is formed by some cellular proteins and up to 16 viral polypeptides, including the RNA dependent RNA polymerase (RdRp or nsp12), the RNA helicase (nsp13) and proteases derived from the proteolytic cleavage of the polyprotein encoded by the viral ORF1ab (Figure 4, step 3). Then, viral RNA replication (Figure 4, step 4) and transcription (Figure 4, step 5) take place attached to cytoplasmic membranes and involve coordinated processes of both continuous and discontinuous RNA synthesis complementary to the +gRNA that produces both −gRNAs and −sgRNAs. New copies of +gRNAs (replication) and +sgRNAs (transcription) are produced using the minority −gRNAs and −sgRNAs as intermediate templates. While +gRNA functions as mRNA for the synthesis of nsps encoded by the ORF1ab, +sgRNAs serve as mRNAs for the translation of structural and accessory genes encoded downstream of the replicase polyproteins. All +sgRNAs are 3′ co-terminal with the full-length +gRNA and thus form a set of nested RNAs. Structural proteins S, E, and M are translated from +sgRNAs (mRNAs) and inserted into the endoplasmic reticulum (ER) (Figure 4, step 6). These proteins move along the secretory pathway into the endoplasmic reticulum-Golgi intermediate compartment (ERGIC) where the viral +gRNAs that are encapsidated by the N protein bud into the membrane resulting in formation of the new mature virus particles (Figure 4, step 7). Following assembly, virions are transported to the cell surface in vesicles and released by exocytosis (Figure 4, step 8). This viral replication cycle consists of several steps involving different RNA molecules functioning as templates for translation and/or RNA dependent RNA synthesis. All of these steps may be sensitive to regulation by RNA secondary structures such as G4s, which may be modulated by viral and/or cellular G4 interacting proteins such as CNBP.

Our bioinformatic analysis revealed that several PQSs are found in both +gRNA and −gRNA, and some of them were selected based on the scores retrieved from several PQSs predictors, including some specially designed for the identification of RNA G4s. From the five selected PQSs, four of them, two from the +gRNA and two from the −gRNA, fold in vitro as stable G4s. Both selected PQSs from the +gRNA that fold as G4 in vitro overlap with the ORF1ab. The PQS +644 overlaps with the region coding the nsp1, which promotes cellular mRNA degradation and blocks host cell translation, resulting in innate immune response blockage. The PQS +3467 overlaps with the region coding the nsp3, a large transmembrane protein comprising several different domains whose precise functions have not been entirely clarified yet. Nsp3 contains the SARS Unique Domain (SUD) that deserves special attention since it is present only in SARS-type coronaviruses and has been associated with the increased pathogenicity of this viral family [69]. Interestingly, SUD has been shown to bind G4s. However, PQS +3467 does not overlap with the SUD coding region. G4s located in the ORF of mRNAs may reduce protein expression levels by acting as roadblocks to the ribosome [70]. Therefore, the ORF1ab translation by cellular ribosomes may be regulated by the +644 and +3467 G4s, especially if they are stabilized by G4 ligands with interesting potential as antiviral compounds. In fact, this translation blocking effect of viral G4s has been proposed for the PQS +28,903, which did not show G4 folding in our assay conditions but was demonstrated to fold as G4 and inhibit the translation of a reporter gene [36] and of the N protein [35] in living cells. Similar functions of RNA G4s were also reported in studies of some other viruses [43,71].

G4s not only act as roadblocks for ribosomes, but also disturb the progression of DNA polymerase [72], RNA polymerase [73], and reverse transcriptase [74], which show processive movement on template nucleotides and should unwind the G4s to continue their reactions. Similarly, RdRp activity could also be inhibited by G4s present in the template RNA. Therefore, the G4s formed in +gRNA (e.g., +644 and +3467 selected and characterized in this work) may act as regulator roadblocks for RdRp catalyzed synthesis of −gRNA. G4s formed in +gRNA could also interfere with RdRp catalyzed synthesis of −sgRNAs intermediates, probably for PQSs other than those selected in this work and overlapped with structural proteins coding region. Moreover, the G4s formed in the −gRNA (e.g., −23,877 and −13,963, both characterized in this work and overlapping the ORF1ab) may act as regulator roadblocks for RdRp catalyzed synthesis of new copies of +gRNA during replication or the synthesis of +sgRNAs during transcription (in the case of other PQSs not selected in this work and overlapped with structural proteins coding region). Although G4 blockage of RdRp has not been experimentally established, it was reported that a stable G4 located at the 3′ end of the hepatitis C virus negative-sense strand could inhibit the RNA synthesis by reducing the RdRp activity [75]. These observations position the PQSs found in the −gRNA as additional putative targets for exploring antiviral strategies targeted to increase (or decrease) the stabilities of viral G4s. However, G4s may not only act as negative elements in nucleic acids metabolism, since there are evidences that these structures may have positive effects in transcription and translation [76], by acting as specific anchoring sites for protein factors or by competing with alternative nucleic acid structures with inhibitory effects. In addition, G4s could be thought as specific anchoring elements for viral RNA encapsidation by structural proteins.

Virus–host cell interplay may involve viral proteins interacting with viral and host G4s, as well as cellular proteins interacting with viral G4s. Recent reports about the specific interaction of viral proteins with SARS-CoV-2 G4s support the relevance of these structures for viral replication. These viral proteins may target not only viral but also cellular G4s. For instance, SUD of nsp3 in SARS-CoV has been shown to bind G4 through a specific macrodomain [69], which is essential for the activity of the RTC of this virus [77]. SARS-CoV SUD binding to viral and/or cellular RNAs with G4s could affect their stability and translation, thus controlling the host cell’s response to the viral infection [69]. Nsp3 of SARS-CoV-2 contains a similar SUD predicted to conserve critical amino acids for G4 binding [32,33,36,78], thus probably sharing with SARS-CoV the viral pathogenic mechanism dependent on nsp3-SUD interaction with G4 structures. Another viral protein, the nsp13 with RNA helicase activity, was also informed as able to bind and probably unfold viral RNA G4s, thus favoring viral translation, transcription, and replication processes [30]. Of notice, besides serving as potential targets for antiviral treatment against SARS-CoV-2, RNA G4s could also be used for the design of biosensors in the detection of viral particles through G4 interaction proteins of SARS-CoV-2 and other coronaviruses, with the potential to replace the antibody-based detection methods and to improve the diagnosis of SARS-CoV-2 and other coronaviruses [79].

With a focus on host proteins, a cellular RNA helicase (Asp-Glu-Ala-Asp (DEAD)-box polypeptide 5 or DDX5) was detected to interact with nsp13 of SARS-CoV, and viral replication was significantly inhibited by knocking down the expression of DDX5 [80]. This suggests that host DDX5 or other DEAD-box helicases could be hijacked by CoVs to enhance the transcription and proliferation of viral genome through G4 unfolding. Other cellular host RNA-binding proteins (SRSF7, HNRNPA1 and TRA2A) were proposed from a predictive analysis as promising candidates for binding and resolving G4s formed in SARS-CoV-2 RNA genome [33]. The recent report about CNBP as the main host cellular protein interacting with SARS-CoV-2 genome, the observation that CNBP expression is induced in response to host cells infection [19], and the fact that CNBP knock-out improves virus-induced cell death [62], highlights the pivotal role of CNBP as an important host protein for controlling viral infection. CNBP is involved in the transcriptional activation of pro-immflamatory cytokines required for activating and maintaining the immune response [67,68], probably acting as a SARS-CoV-2 antiviral regulator [19]. In addition, CNBP is a single-stranded nucleic acid binding protein with chaperone activity that may control gene expression (at transcriptional and translational levels) through G4 unfolding [65,66]. Considering these evidences, together with the variety of putative functions of G4s in SARS-CoV-2 replication cycle and the G4 unfolding capacity of CNBP showed here, we hypothesize that CNBP may act as a direct regulator of viral gene expression, transcription, and replication, adding relevant evidences for understanding the role of this protein in the control of SARS-CoV-2 infection. Although previous data has shown that CNBP binds to +gRNA within infected cells and there is still no evidence of interaction with −gRNA or −sgRNAs, our results show for the first time that CNBP is capable of binding and unfolding in vitro G4s present in the −gRNA. This opens the hypothesis that this protein may interact with both viral RNA strands, favoring the unwinding of G4s and controlling viral replication, transcription, and gene expression in those steps where G4 structures may be acting. To date, no pharmacological products with therapeutic potential have been designed for modifying CNBP activity. Future experimental work is needed for assessing the function of G4s present in +gRNA and −gRNA in viral host cell infection, as well as the action of cellular proteins such as CNBP in these processes, with perspectives of understanding useful molecular mechanisms for the design of new antiviral strategies.

## 3. Materials and Methods

### 3.1. Bioinformatics

The linear genomes of the six viruses used here were downloaded from the genome database of the National Center for Biotechnology Information (NCBI, https://www.ncbi.nlm.nih.gov/, accessed on 2 January 2021) [81]. Full names and accession numbers are indicated in Table 1.

PQSs prediction was performed by the web-based server QGRS mapper (https://bioinformatics.ramapo.edu/QGRS/index.php, accessed on 2 January 2021) [45]. PQSs found by QGRS mapper were analyzed also by PQS Finder [46] and G4RNA screener [47]. G4RNA screener is a web algorithm that identifies regions in RNA sequences prone to fold into G4 based on three scoring systems: cGcC (Consecutive G over consecutive C ratio) that addresses the issue of competition in between G4 and Watson-Crick structures [48], G4H (G4Hunter) which is similar to the cGcC but was built to analyze DNA sequences [49], and G4NN (G4 Neural Network) that is based on sequences of the G4RNA database converted into vectors of their trinucleotide content to train an artificial neural network [50]. The parameters used for the algorithms are detailed in Appendix A.

The nucleotide and amino acid variations of SARS-CoV-2 genome and their associated data were searched by using the Nextstrain tool for analysis and visualization of 325,005 SARS-CoV-2 full genomic sequences sampled by different laboratories worldwide between 10 January 2020 and 11 January 2021, available in GISAID database (https://www.gisaid.org/, accessed on 11 January 2021) [52].

The predictions of RNA secondary structures were performed using the web based software RNAfold (http://rna.tbi.univie.ac.at/cgi-bin/RNAWebSuite/RNAfold.cgi, accessed on 12 February 2021) [82], and NUPACK (http://www.nupack.org/, accessed on 12 February 2021) [83], using the settings indicated in Appendix A.

### 3.2. Oligonucleotides

Synthetic single-stranded desalted oligoribonucleotides (Table 2) were purchased from Sigma-Aldrich, dissolved in bidistilled water and stored at −20 °C until use. Concentrations were determined by spectrophotometry using extinction coefficients provided by the manufacturer.

### 3.3. Circular Dichroism (CD) Spectroscopy

Intramolecular G4s were folded by dissolving 2 μM RNA oligonucleotides in 10 mM Tris-HCl pH 7.5 and different KCl or LiCl concentrations, as indicated in each figure, heating for 5 min at 95 °C and slowly cooling to 20 °C. For analysis of CNBP and BSA effect, prior to CD spectroscopy, proteins dissolved in CNBP buffer were added to the pre-folded G4 (8 μM) at equimolar concentrations and incubated at 37 °C for 30 min. CD spectra were recorded at 20 °C over a wavelength range of 220–320 nm with a Jasco-1500 spectropolarimeter (10 mm quartz cell, 100 nm/min scanning speed, 1 s response time, 1 nm data pitch, 1 nm band width, average of four scans). The spectral contribution of buffers, salts, and proteins were appropriately subtracted by using the software supplied with the spectropolarimeter. Three qualitative rules-of-thumb exist for CD spectral features associated with particular G4 topologies, namely parallel (with an important positive band around 264 nm and a relative shallow negative band at 245 nm), antiparallel (with a positive band around 295 nm, and a negative band around 265 nm), or hybrid (with two positive bands around 295 nm and 264 nm, and a negative band around 245 nm) [84,85]. The CD melting curves were recorded by ellipticity measurements between 20 °C and 95 °C at the wavelength corresponding to the maximum observed at the initial temperature (20 °C) for positive band around 264 nm, using the same parameters set for the spectra, except for 5 nm band width, a temperature increase speed of 1 °C/min, and a sampling interval of 0.1 °C. Data was analyzed in SigmaPlot 11.0 with a nonlinear least squares fitting procedure assuming a two-state transition of a monomer from a folded (G4) to an unfolded state with no change in heat capacity upon unfolding [86,87]. Melting curves were plotted as the fractional population of the G4-folded oligonucleotides (FG4 = [θ(T) − θU]/[θG4 − θu]) vs. temperature, where θf and θu are the ellipticities of the fully folded and unfolded states, respectively. The reported Tm represent the temperature at which both states are equally populated (FG4 = Fu = 0.5). Tm, ΔH, θf, and θu are those which provide the best fit of experimental melting data and the shown spectra and melting curves are representative of at least three independent experiments. ΔG_37 °C_ were estimated following the procedures indicated elsewhere [86].

### 3.4. 1D ^1^H Nuclear Magnetic Resonance (NMR)

NMR spectroscopy provides information about the type of base associations in the nucleic acid oligonucleotides by imino protons signals in the spectral region between 9 and 16 ppm. For instance, Watson-Crick typically presents signals clustered around 13–14 ppm, G4 around 11–12 ppm and i-motifs around 15–16 ppm [60]. In this work, NMR spectra were acquired at 20 °C on a 700MHz Bruker Avance III spectrometer (Bruker Biospin, MA, USA) equipped with a triple resonance inverse NMR probe (5 mm ^1^H/D-^13^C/^15^N TXI). Samples contained 50 µM RNA oligonucleotides folded in 10 mM Tris at pH 7.5 supplemented with 100 or 200 mM KCl as described for CD spectroscopy. We used 5 mm Shigemi tubes that were previously treated with HCl 1M and washed extensively with water and ethanol to remove RNAses. 1D ^1^H NMR spectra were registered using a pulse sequence with excitation sculpting (zgesgp) for water suppression [88]. We used 8K points, 4096 scans, a recycling delay of 1.4 s and a sweep width of 22 ppm. Experimental time for each NMR spectrum was 1 h 56 min. We repeated the spectra at different time points to discard degradation of the RNA oligonucleotides. Processing was done using an exponential window function multiplication with a line broadening of 10 Hz and baseline correction. We used Topspin 3.5 software (Bruker, Biospin, MA, USA) for acquisitions, processing, and analysis of the NMR spectra.

### 3.5. Thermal Difference Spectroscopy (TDS)

Two μM RNA oligonucleotides folded in 10 mM Tris at pH 7.5 supplemented with 100 or 200 mM KCl as described for CD spectroscopy were scanned to measure absorbance over the wavelength range of 220−320 nm using a scan speed of 100 nm/min and a data interval of 1 nm and a 10 mm quartz cell. Spectra were recorded at 20 °C and then at 70 °C using a Jasco V-630BIO spectrophotometer with peltier temperature control. The absorbance spectra obtained at these two temperatures were subtracted (A 70 °C–A 20 °C) and plotted on a graph to obtain TDS according to [61]. Typical G4 signature presents two positive peaks around 243 and 273 nm and a negative peak at 295 nm.

### 3.6. ThT Fluorescence Assays

ThT (3,6-Dimethyl-2-(4-dimethylaminophenyl) benzothiazolium cation, Sigma-Aldrich T3516) fluorescence assays were performed as previously described [89]. Briefly, 100 μL of 2 μM RNA oligonucleotides folded in 10 mM Tris at pH 7.5 supplemented with 100 or 200 mM KCl as described for CD spectroscopy were mixed with 100 μL of 1 μM ThT and loaded into 96-well black microplates (Greiner, NC, USA). Fluorescence emission measurements were performed using a microplate reader (Synergy 2 Multi-Mode Microplate Reader, BioTek, VT, USA) with excitation filter of 485 ± 20 nm and an emission filter of 528 ± 20 nm. Each sample was tested by triplicate and fluorescence values were relativized to ThT fluorescence in the absence of oligonucleotides (F_0_). A threshold of 10-fold increase was used for considering G4 formation. NRAS RNA oligonucleotide representing a PQS from human NRAS 5′ UTR [90] was used as positive control for the assay (Appendix A). 

### 3.7. CNBP Expression and Purification

The pET-32a-TEV-CNBP plasmid [66] was used for recombinant expression and purification of tag-free human CNBP following guidelines detailed elsewhere [91]. CNBP was obtained in CNBP buffer (50 mM Tris–HCl pH 7.5; 300 mM NaCl; 1 mM DTT, 5 mM Imidazole and 0.1 mM ZnCl_2_), which was used in several in vitro assays as a control.

### 3.8. Electrophoretic Mobility Shift Assay (EMSA)

EMSAs were performed as described previously [92] with some modifications for non-radioactive detection. Binding reactions were performed in 20 mM HEPES pH 8.0, 10 mM MgCl_2_, 1 mM EDTA, 0.5 µg/µL Heparin, 1 mM DTT, 1 µg/µL BSA, and 10% glycerol. Probes were added to a final concentration of 0.75 µM. 100 mM KCl or LiCl were added, depending on the folding condition of the probe, as indicated in Figure 3a. Final reaction volumes were 20 µL. Binding reactions were incubated for 30 min at 37 °C and then loaded in 14% polyacrylamide gels containing 5% glycerol in TBE 0.5X. After electrophoresis, gels were exposed for 10 min to SYBR Gold stain [93] to detect bands, the fluorescence of which was subsequently registered in a Typhoon FLA 7000 Scanner (GE Healthcare, NJ, USA) using ImageQuant 5.2 software. RNA oligonucleotides used as probes were previously folded by thermal denaturation and slow renaturation in a buffer (10 mM Tris-HCl pH 8.0, 1 mM EDTA) in the presence of LiCl or KCl at concentrations of 100 or 200 mM, as indicated for each case in the figure caption (Figure 3). The CNBP buffer was used for the CNBP dilutions and for the reactions with no added CNBP.

## Figures and Tables

**Figure 1 ijms-22-02614-f001:**
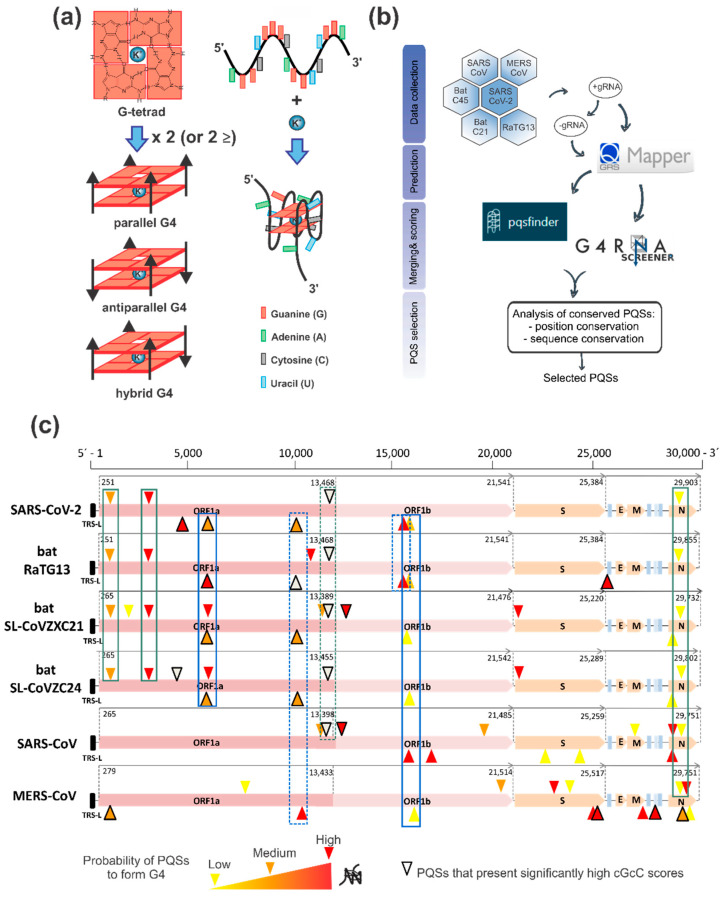
Identification and selection of PQSs in the genomes of SARS-CoV-2 and other members of the *Coronaviridae* family. (**a**) Cartoon representing the formation of G4s. Left: G-tetrads are formed by the planar arrange of four G nucleobases stabilized by lateral Hoogsteen-type hydrogen bonds. The stacking of two or more tetrads and the coordination of K^+^ form the G4 structure. Depending on the relative orientation of the G-tracts, G4s may be parallel, antiparallel, or hybrid. Right: G4 formation by the folding on itself of a G-rich RNA strand with at least four contiguous G-tracts interspersed with short nucleotide loops. The formation of a parallel G4 is represented. (**b**) Schematic summary of the bioinformatic workflow conducted for PQSs identification and selection in the positive- and negative-sense RNA genomes of SARS-CoV-2, RaTG13, bat-SL-CoVZC45, bat-SL-CoVZXC21, SARS-CoV, and MERS-CoV. (**c**) Schematic representation of the location of the selected PQSs in the analyzed genomes. The organization of the coding regions for the main viral proteins are represented for each virus: ORF1a (open-reading frame 1a), ORF1b (open-reading frame 1b), S (spike), E (envelope), M (membrane) and N (nucleocapsid), black rectangle represents transcription-regulatory sequences (TRS) in the 5′ untranslated region (UTR). PQSs found in the positive-sense strand are represented above the genome while PQSs found on the negative-sense strand represented below the genome. PQSs are represented with the following color code: PQSs with high probability to form G4 (red), PQSs with medium probability to form G4 (orange) and PQSs with low probability to form G4 (yellow). PQSs that present significantly high cGcC scores (>150) are highlighted using thick edges. PQSs conserved in position and sequence are indicated with solid green line boxes (for positive-sense strand) and solid blue line boxes (for negative-sense strand). Dashed lines indicate PQSs conserved in position that were not selected due to lack of sequence conservation, to conservation in less than four genomes or to not overpassing the selection criterion.

**Figure 2 ijms-22-02614-f002:**
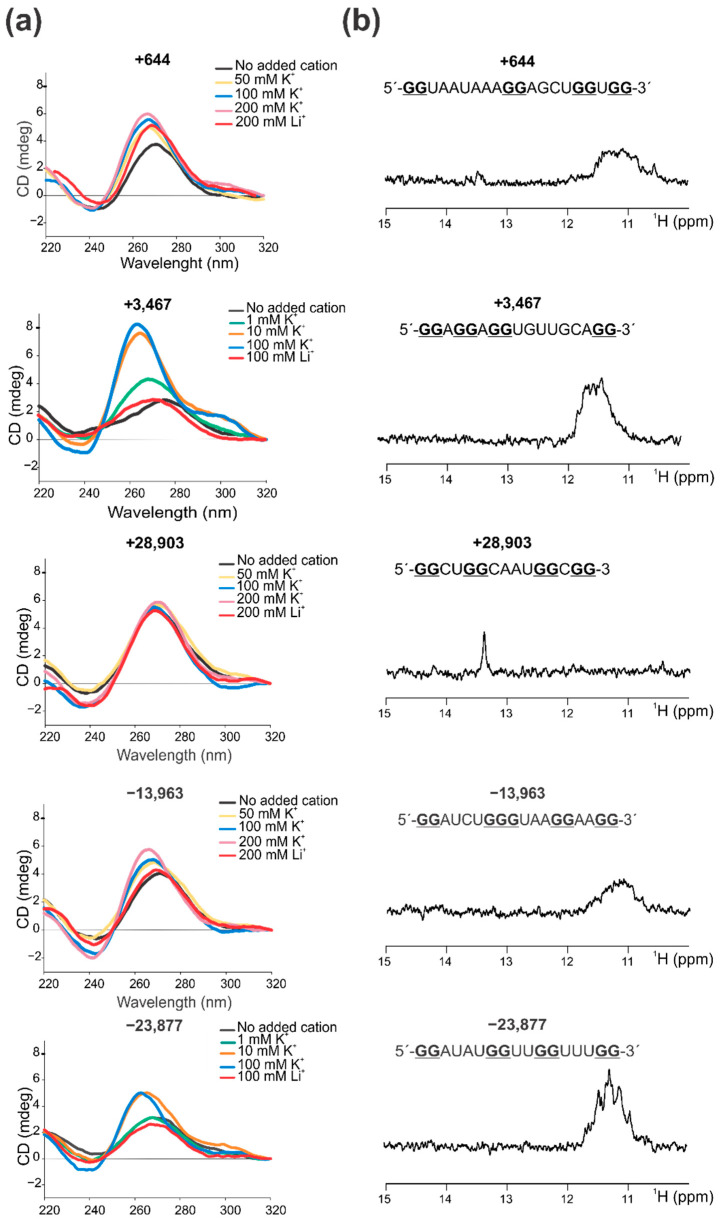
Evidence by CD and NMR spectroscopy of the in vitro G4 structures formed by the selected PQSs. (**a**) CD spectra were obtained for each RNA sequence (named by the PQS position) folded in the absence and in the presence of increasing K^+^ concentrations, or in the presence of Li^+^ at the highest concentration used for K^+^. Concentrations are indicated for each plot. (**b**) 1D ^1^H NMR spectra obtained for each RNA sequence (named by the PQS position) folded in the presence of K^+^ at the highest concentration used for CD. RNA sequence for each PQS are represented above NMR spectra, and guanine nucleotides predicted to participate in the G4 formation are indicated in bold and underlined.

**Figure 3 ijms-22-02614-f003:**
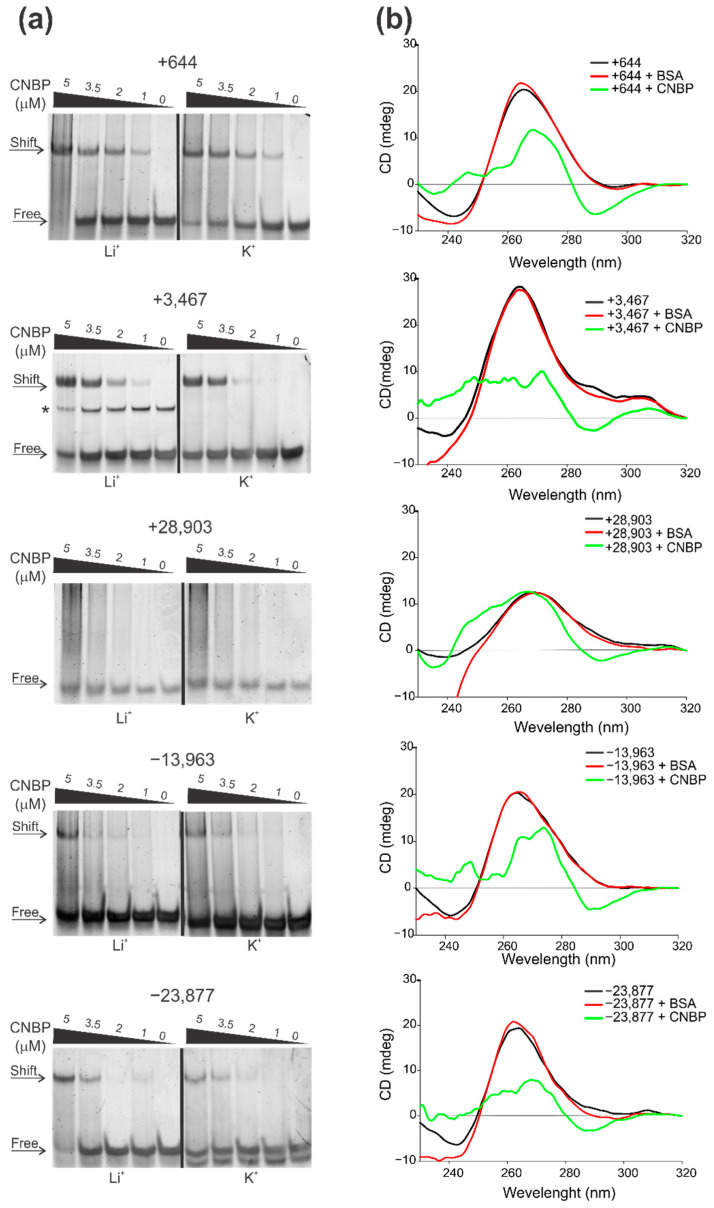
CNBP binds and unfolds the G4s formed by PQSs in the +gRNA and −gRNA of SARS-CoV-2. (**a**) Representative EMSAs performed using PQSs (named by the PQS position) as probes folded in the presence of Li^+^ (left) or K^+^ (right) and then incubated in the absence or presence of increasing concentrations of CNBP. Free and shifted probes are indicated by arrows at the left of the gels. The +3467 probe folded in the presence of Li^+^ presents a minority band (marked with *) of lower mobility probably due to a self-assembled dimeric or multimeric complex. (**b**) CD spectra obtained for 8 µM oligonucleotides (named by the PQS position) folded as G4 in the presence of K^+^ and incubated in the absence of protein or in the presence of CNBP (1:1 molar ratio) or BSA. For EMSAs and CD, K^+^ and Li^+^ concentrations used for G4 folding were 100 mM (+3467 and −23,877) or 200 mM (+644, +28,903 and −13,963).

**Figure 4 ijms-22-02614-f004:**
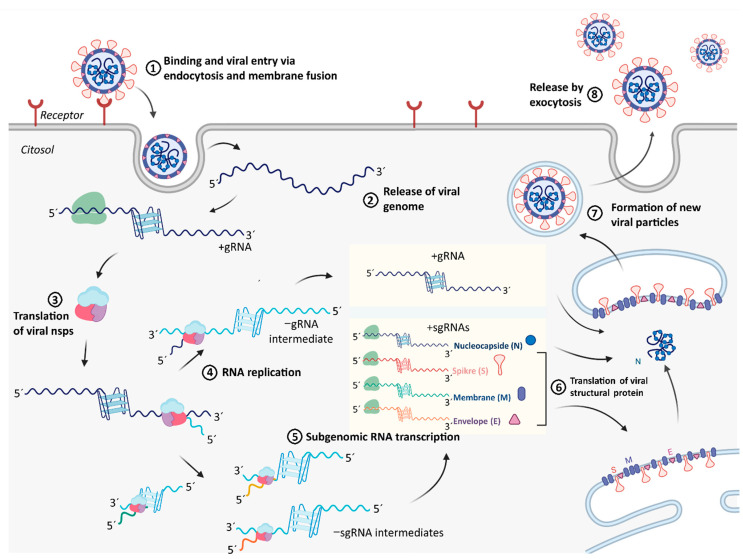
Possible role of G4s in the SARS-CoV-2 and coronaviruses replication cycle. Different steps of viral infection and replicative cycle are detailed: (1) binding to host cell receptor, (2) entry to host cell, (3) translation of viral nsps from ORF1ab, (4) viral RNA replication, (5) viral RNA transcription, (6) viral structural proteins translation, (7) encapsidation of viral +gRNA and formation of mature virus particles, (8) viral release. Viral G4s and the G4-interacting proteins, such as CNBP, may participate in the regulation of the efficiency of steps 3–7, which could be targets of drug candidates for antiviral therapies. Created with BioRender.com.

**Table 1 ijms-22-02614-t001:** Numbers of found and selected PQSs for +gRNA and −gRNA of the analyzed viruses.

Betacoronavirus	Genome Accession Number	% G	Number of PQSs in QGRS Mapper	Selected PQSs				% of Selected over Predicted PQSs
				Low Probability to Form G4	Medium Probability to Form G4	High Probability to Form G4	Total	
+gRNA								
**SARS-CoV-2**	NC_045512.2	19.6	37	1	1	1	3	8.1
**RaTG13**	MN996532.2	19.6	37	1	2	1	4	10.8
**Bat-SL-CoVZXC21**	MG772934.1	20.1	32	2	2	4	8	25.0
**Bat-SL-CoVZC45**	MG772933.1	20.2	29	1	1	3	5	17.2
**SARS-CoV**	NC_004718.3	20.8	49	2	2	2	6	12.2
**MERS-CoV**	NC_019843.3	20.9	40	3	1	2	6	15.0
−gRNA								
**SARS-CoV-2**	NC_045512.2	18.4	19	-	3	2	5	26.3
**RaTG13**	MN996532.2	18.4	20	-	1	3	4	20.0
**Bat-SL-CoVZXC21**	MG772934.1	18.7	18	2	2	-	4	22.2
**Bat-SL-CoVZC45**	MG772933.1	18.7	16	2	2	-	4	25.0
**SARS-CoV**	NC_004718.3	20.0	29	2	-	3	5	17.2
**MERS-CoV**	NC_019843.3	20.3	38	2	2	5	9	23.7

**Table 2 ijms-22-02614-t002:** Information of selected SARS-CoV-2 PQSs.

PQS Name	Genome	Length	Sequence	Prediction Scores					Reference of Previous Prediction	Reference of Experimental Evidence of G4 Formation
				G4RNA Screener			QGRS Mapper	PQS Finder		
				cGcC	G4H	G4NN				
+644	ORF1ab nsp1	20	**GG**UAAUAAA**GG**AGCU**GG**U**GG** **C** **G** **G** **C**	17	0.8	0.69	30	20	[30,31,32,33,34,36]	[36]
+3467	ORF1ab nsp3	17	**GG**A**GG**A**GG**UGUUGCA**GG**A **A**** ****A**** ****G**** ****A**** ****A**** ****G****AA****G** **U**** ****U**	18	1	0.87	30	24	[30,31,32,33,34,36]	[34]
+28,903	N	15	**GG**CU**GG**CAAU**GG**C**GG** **UAU** **UU** **UU** **UU****U****AU** **AU** **--****-****--** **C **	5.33	0.87	0.57	33	27	[30,31,32,33,34,36]	[34,35,36]
−13,963	ORF1ab nsp12	18	**GG** AUCU**GGG**UAA**GG**AA**GG** ** ** **G** ** ** **G** ** ** **G** ** ** **AA**	22	1.11	0.97	34	23	[32,33,36]	-
−23,877	ORF1ab nsp3	17	**GG** AUAU**GG**UU**GG**UUU**GG** **AA** ** ** **C** ** ** **C** ** ** **A** ** ** **A**	160	0.94	0.02	34	24	[32,33,36]	-

Note: nucleotide variations indicated below each PQS are highlighted according to their consequence to disrupt G-tracts and impede PQSs or reduce PQSs scores (red), do not affect G-tracts and may be neutral for PQSs (yellow) or produce G-tracts extension and increase PQSs scores (green). Dashes indicate single nucleotide deletions and are highlighted with the same color code as substitutions in respect to their effect on PQSs.

## Data Availability

The data presented in this study are available in the article and Appendix A.

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
