# Peer review of "CNBP Binds and Unfolds In Vitro G-Quadruplexes Formed in the SARS-CoV-2 Positive and Negative Genome Strands"

_ijms, 2021, doi:10.3390/ijms22052614_

Round 1

Reviewer 1 Report

The presented manuscript by Bezzi et al. is very well-written and gives us a comprehensive and conclusive picture of PQSs and selected in vitro validated G4s in the genomes of SARS-CoV-2 and its close relatives. The added value of this work (in comparison with the previously published early reports) lies mainly in:

1.) proposing a novel method of classifying SARS-CoV-2 PQSs

2.) carefully designed experiments and hypothesis

3.) in vitro validation of 2 SARS-CoV-2 PQSs located on the negative-sense RNA

4.) the analysis of predicted PQSs in regards to GISAID mutations

5.) in vitro validation of human protein CNBP that unfolds SARS-CoV-2 G4s

6.) extensive Discussion of the obtained results and a nice final scheme

I have only a few minor points and suggestions (please, see below)

Minor points:

In the title, there is a typo ... "fomed" ---> formed

In the Abstract:

has become has ... redundant "has"

negative-sense replicative intermediaries ... should be "intermediates"

Consider using Italics, when referring to the taxonomical units (e.g. Coronaviridae family)

line 48: "Instead, the other three human betacoronaviruses are more recent and rare" ... please, reformulate it, unfortunately, SARS-CoV-2 is no "rare" nowadays

line 54: the closest relative of SARS-CoV-2 is RatG13 bat coronavirus, not?

line 60: "is released ready to be translated" ...some word is redundant here

line 64: "intermediaries" ---> intermediates (please, fix in the whole manuscript)

Figure 1a ... typo in "Uracyl" (should be Uracil)

line 85 ... should be "guanosine nucleosides" or "guanine nucleotides", not "guanosine nucleotides"

line 86 ... "nucleotidic" sequences (?) - and also line 108

line 95 ... add also "Archaea" to the sentence "Besides in human beings, G4s are widespread across nucleic acids of all the taxonomic phyla including bacteria and viruses" altogether with this recent reference

line 114 ... "(CNBP), the main cellular protein 114 bound to SARS-CoV-2 RNA genome in infected human cells" ... quotation for this statement needed

lines 167-177 are very difficult to follow, please, consider moving this section to the Methods or reformulate it to be more "reader-friendly"

line 195 ... "ORF1 a" and "ORF1 b" should be written without spaces (ORF1a)

figure 1c ... the scale bar of length 29903 nt is valid only for SARS-CoV-2 (e.g. MERS-CoV is of length 30119 nt)

lines 228 and 237... what is "PG4s"? should be "PQSs (?)

line 240 ... "may be not be"

Table 2 ... please explain the meaning of black, red, and yellow horizontal lines (and also solid and dashed ones) in the note below the Table. Also, from the title of this Table, it is not clear, if these are SARS-CoV-2 PQSs, or conserved PQSs in all five selected viruses

Line 297 ... "On the other hand, not conserved PQSs that are unique for a particular virus may 297 also play a central role in the ability of the virus to adapt to new environmental challenges 298 and infect and replicate in novel hosts."  should be nonconserved

Line 320 ... "Li+, which is a G4-non-stabilizing cation" is a somewhat controversial statement and should be reformulated

Figure 2a ... y-axes [mdeg] should be scaled the same, i.e. -4 to 10, to be comparable (?)

Lines 346 - 356 ... could authors check, if the +28903 sequence has the potential to form stable Watson-Crick base pairs, as they suggested? E.g. by some primer-dimer or RNA fold prediction tool? Just a suggestion, not a critical point

Line 363 ... in vitro should be in Italics, please check elsewhere

Line 422 ... "PQSs folded (in presence of K+) or unfolded (in presence of Li+) as G4s" should be reformulated

Figure 3b ... consider rescaling the y-axes to be the same (comparable)

Line 455 ... CNBP should be in italics (?) (you are referred to a gene (?)

Line 505 ... I suppose that figure was "Created in Biorender.com". This statement should be added here. Also, the resolution of the picture is not very high, Biorender offer high-resolution export option, not? It is a nice and comprehensive scheme, so it would be worth of it...

Line 513 ... "The PQS +3467 overlaps with the region coding the nsp3, a large trans-513 membrane protein comprising several different domains whose precise functions have 514 not been entirely clarified yet" ... nsp3 contains G4-interacting domain (SUD M domain), maybe you wish to mention it here...in addition, does the +3467 PQS span SUD M domain of nsp3

Line 606 ... link to QGRS mapper is nonfunctional/broken (should be written without "www" (?))

Line 610 ... aminoacidic, nucleotidic

Line 618 ... please, add details about the purity of RNA oligonucleotides (e.g. desalted; HPLC purified; unpurified,...)

Line 670... typo "ptesents" should be "presents"

Reviewer 2 Report

In this manuscript entitled as “CNBP binds and unfolds in vitro G-quadruplexes fomed in the SARS-CoV-2 positive and negative genome strands”, Georgina Bezzi et al. identified some PQSs in the positive and negative strand of SARS-CoV-2 and the related viruses. The authors utilized the known PQSs prediction programs which are opened in web sites. The positive strand has the PQSs as reported in the previous studies. Moreover, the authors found that the negative strand also had the PQSs. It was confirmed with spectroscopic methods that some of the PQSs actually folded into G-quadruplex structures.  Finally, the authors find that CNBP bound and unfolded the G-quadruplexes. The some of the findings in this study are consistent with the previous publications, meaning that the results are already known. However, there are additional new results in this manuscript. The results and the conclusions shown in this manuscript are emergent topics, thus this reviewer can suggest publication of this manuscript in the IJMS with some additional results. The comments are the followings:

  1. Please provide results for thermal profile of the G-quadruplex structures found in this study. Thermal melting analysis traced with UV absorption at 260 nm and 295 nm could be useful information for understanding how stable the G-quadruplexes.
  2. More evidences to show formation of the G-quadruplexes are required. For example, ligands such as NMM and ThT which emit bright fluorescence are useful for the PQSs, and for full-length RNAs. Since C-rich regions nearby the G-rich PQSs potentially compete with the G-quadruplex formation due to the stem-loop duplex formation, the G-quadruplex formation could be confirmed with the wider region with nearby sequences.
  3. Please provide Kd values of VNBP with the target G-quadruplexes. And also provide thermal stability of the target G-quadruplexes in the presence of CNPB.
  4. Figure 4: The roles of G-quadruplexes shown in Figure 4 are totally schematic without any evidence. Can the authors provide some additional results for the reverse transcription, replication, and translation with the RNAs with or without the PQSs?

Reviewer 3 Report

The manuscript describes the study of the interaction between G-quadruplex structures identified within SARS-CoV-2 and other viruses and the CNBP protein. My comments are limited to the spectroscopic and thermodynamic characterization of these RNA structures.

- There is a spelling mistake in the title (“fomed”)

- The description of the CD measurements and melting experiments in pages 17-18 does not include the heating rate at which melting experiments were done. On the other hand, the plots of folded fraction vs. T (supplementary figure S3) show a huge variability, which is not acceptable. In addition, some experimental data seem not to fit well the proposed curves. I think that these experiments should be repeated at a lower heating rate and taking care of the instrumental setup (band width?, response?, scanning speed?, data pitch?, formation of bubbles?,…) to prevent that high variability.

- In addition to the determined Tm values, the stability of folded nucleic acid structures should be discussed in terms of thermodynamic parameters (for instance, change of free Gibbs energy at 37oC). I would suggest the inclusion of a simple thermodynamic analysis from the plots of folded fraction vs. T, considering a two-state process. I think that this analysis could reinforce the discussion about the stability of the structures formed by some sequences, such as +28903, which seems to form a less stable structure (probably a hairpin, according to Nupack analysis) than the others (e.g., the G-quadruplex formed by + 3467).

- CD spectra of RNA and CNBP (1:1 mixtures) in Figure 3 show shapes and intensities quite different from those observed for RNA or even RNA:BSA mixtures. I wonder whether this could be related to precipitation and/or to a high value of absorbance. Did the authors check the presence of these potential artifacts?

Round 2

Reviewer 3 Report

The authors have answered all my comments and suggestions. Therefore, I suggest the acceptance of this manuscript.